# Event dependent overall survival in the population-based LIFE-Adult-Study

**Samira Zeynalova**[1]*, **Katja Rillich**[1], **Eike Linnebank**[1], **Tina Stegmann**[2], **Michael Brosig**[3], **Matthias Reusche**[1,4], **Markus Loeffler**[1,4]

1 Institute for Medical Informatics, Statistics, and Epidemiology, University of Leipzig, Leipzig, Germany, 2 University Clinic Leipzig, Leipzig, Germany, 3 Academic Teaching Practice, University of Leipzig, Leipzig, Germany, 4 LIFE—Leipzig Research Centre for Civilization Diseases, University of Leipzig, Leipzig, Germany

* Samira.Zeynalova@imise.uni-leipzig.de

## Abstract

### Backround

Information about the direct comparability of big data of epidemiological cohort studies and the general population still is lacking, especially regarding all-cause mortality rates. The aim of this study was to investigate the overall survival and the influence of several diagnoses in the medical history on survival time, adjusted to common risk factors in a populations-based cohort.

### Methods

From 10,000 subjects of the population-based cohort LIFE-Adult-Study (Leipzig Research Centre for Civilization Diseases), the medical history and typical risk factors such as age, smoking status and body-mass-index (BMI) were assessed. The survival status was identified from the saxonian population register. Univariate and multivariate analyses were used to determine the influence of the medical history and risk factors on overall survival. To develope an optimal model, the method by Collet [1] was used.

### Results

The mortality rate of the participants is approximately half the mortality rate expected for the german population. The selection bias in epidemiological studies needs to be considered whenever interpreting results of epidemiological cohort studies. Nevertheless we have shown that several diagnoses proved to have a negative influence on overall survival time even in this relatively healthy cohort. This study showed the significantly increased mortality risk if the following diseases are reported in medical history of the participants in a large *population*-based *cohort* study including *adults aged* 18 and *over*: diabetes mellitus (HR 1.533, p = 0.002), hypertension (HR 1.447, p = 0.005), liver cirrhosis (HR 4.251, p < 0.001), osteoporosis (HR 2.165, p = 0.011), chronic bronchitis (HR 2.179, p < 0.001), peptic ulcer disease (HR 1.531, p = 0.024) and cancer (HR 1.797, p < 0.001). Surprisingly, asthma has the opposite effect on survival time (HR 0.574, p = 0.024), but we believe this may be due to an overrepresentation of mild to moderate asthma and its management, which includes educating patients about a healthy lifestyle.

**Data Availability Statement:** Data cannot be shared publicly because of data protection reasons. Data availability is only possible via a project agreement. Data are available from the LIFE-Adult-

Study Institutional Data Access / Ethics Committee (contact via https://ldp.life.uni-leipzig.de/) for researchers who meet the criteria for access to confidential data.

**Funding:** LIFE was funded by means of the European Union, by the European Regional Development Fund (ERDF) and by funds of the Free State of Saxony within the framework of the excellence initiative (project numbers 713-241202, 713- 241202, 14505/2470, 14575/2470)" and „Data evaluation for this publication was supported by Sanofi Genzyme, Germany, by a non-restricted scientific grant.

**Competing interests:** The authors have declared that no competing interests exist.

## Conclusion

In the LIFE-Adult-Study, common risk factors and several diseases had relevant effect on overall survival. However, selection bias in epidemiological studies needs to be considered whenever interpreting results of epidemiological cohort studies. Nevertheless it was shown that the general cause-and-effect principles also apply in this relatively healthy cohort.

## Introduction

The aim of this study was to investigate the mortality risk among the participants of a large epidemiological cohort study and to determine the influence of the medical history, adjusted by several known risk factors as age, body mass index (BMI) or smoking status, on mortality risk in this population. There are few publications that perform this type of analysis. For example the data of the nurses' health study was used to investigate the influence of several risk factors in life style, as physical activity, smoking, alcohol consumption and nutrition, on mortality [2]. But in most cases, mortality associated with a particular risk factor or disease is determined, e.g. hypertension [3], diabetes mellitus [4] or hypothyroidism [5]. Independent examination of medical history, as it is done here, provides an open-ended approach to examine the general risks of particular diagnoses in medical history in a mixed population.

The analysis presented here was conducted using data collected as part of the LIFE-ADULT-Study. The LIFE-ADULT-Study is a large epidemiological prospective cohort study in which 10,000 randomly selected adults underwent an extensive screening program [6]. Among many other examinations and surveys, data for risk factors such as BMI and smoking were collected, but also extensive data on previous diseases were requested. In this publication, these collected diagnoses from medical history of the participants, adjusted by the risk factors, are associated with the mortality risk in the population examined in the LIFE-ADULT-Study.

The diagnoses investigated in this study mainly focus on chronic diseases or events that are a result of an underlying chronic condition. It is a known fact that chronic diseases cause premature death [7]. This study addresses the question whether this fact can also be shown in this study population.

## Materials and methods

### Study population

The LIFE-ADULT-Study is a population-based cohort study, which randomly selected 10,000 residents aged 18–80 years from Leipzig, Germany. It is part of the "Leipzig Research Center for Diseases of Civilization" (LIFE). The study population was stratified by age and gender as previously described in detail [6].

The study was initiated to assess prevalence and incidence of common diseases and subclinical disease phenotypes, identify early onset markers, genetic predispositions, and the role of lifestyle factors in major civilization diseases. The baseline examination took place from August 2011 until December 2014. All subjects underwent an extensive core assessment program (5–6 h) including structured medical interviews, medical and psychological questionnaires, physical examination, and biospecimen collection [6]. The participation rate, concerning the number of persons invited until reaching the planned sample size, was 33%.

All subjects signed an informed consent form and the study protocol was in accordance with the Declaration of Helsinki and approved by the Leipzig University Ethics Committee.

## Assessment of risk factors

Date of birth, sex, body mass index, smoking status and former diagnoses were assessed at the baseline examination in the study center. Age was calculated at the time point of enrolment in the LIFE-ADULT-Study as difference from current date and date of birth. Body mass index was measured during bioimpedance analysis. The Stadiometer 274 and BIA-scale 515 of SECA were used.

Smoking status and diagnoses were assessed by an interview. If a participant had stopped smoking at baseline, this was rated as a no smoking status. To assess the diagnoses that the participant had received throughout life up to baseline examination, the interviewer asked the participant for each diagnostic complex whether any of the included diseases had ever been diagnosed by a physician. Any "I don't know" response was excluded from the analysis. Particularly cardiovascular-based pathologies and their known risk factors were evaluated (e.g. myocardial infarction, stroke, hypertension, diabetes, etc.).

## Mortality status and objectives

For capturing the events of deaths among the LIFE-ADULT participants in the observed period, we submitted a query to the Saxon Registration Register (→"Sächsisches Melderegister") regarding death events of all 10,000 participants. The timepoint of the occurance of the deaths was used for analysis of primary and secondary objectives of this study: survival time and calculation of the cumulative incidence of mortality from August 2011 (the baseline start) until end of 2019.

The study's primary objective was to determine the overall survival (OS). OS was defined as the time from baseline examination at the study center to death from any cause of death for deceased participants or to the date of search in the registry for non-deceased subjects. OS was the only endpoint in this work, as the causes-of-death are still unknown or undetermined in the LIFE-Adult-Study. The secondary objective of the study was to assess the representativeness of the LIFE cohort by comparing mortality among the population investigated in the LIFE-ADULT-Study with the overall mortality for the total population of Germany.

## Statistical analysis

In univariate analysis, the primary endpoint overall survival (OS) for each diagnosis was estimated using the method of Kaplan and Meyer. The log-rank test was used for comparison.

Multivariate analyses were performed using a Cox proportional regression model adjusting for BMI, age, gender and smoking status.

To identify prognostic factors for death, we used the proportional hazard model. We did not use automatic selection procedures but proceeded in a stepwise approach for including single factors as suggested by Collet (Sauvanet A 2005).

This optimal model determined factors (diagnoses in history) that increase the risk of mortality.

Patient characteristics between groups were compared using chi-square tests and, if required, by Fisher's exact tests for categorical or nonparametric variables. Mann-Whitney U test was used for continuous variables.

The significance level was 5% (two-sided). The relevant factors with p less than 10% were also shown in the results of the multivariate analysis. The statistical software used for analyses was SPSS (version22·0, IBM, Chicago, United States).

The estimated overall mortality was calculated based on the life table of the german population. For every calender year from 2011 to 2019 the probability of death was calculated for the number of participants included in the LIFE-ADULT-Study. Age and sex of the participants

were used to select the probability of death from the life table. The age of the participants was adjusted every year to take into account that the probability of death increases with age.

## Results

### Participant characteristics

All 10,000 subjects were included into analysis regarding the baseline characteristics (Table 1). The baseline characteristics were compared between deceased and non-deceased subjects within the anlysed period. Median age was 57.9 (47.6; 68.2) years. 52% (n = 5197) were women. Median BMI was 26.6 kg/m$^2$ (23.9; 30) and 22% (n = 2052) of participants smoked at the time of inclusion in the study.

We also characterised the collective by looking at the distribution of the diseases recorded, particularly cardiovascular-based pathologies and their known risk factors. Total numbers and proportions of pathologies regarding the cardiovascular system are:

**Table 1. Baseline characteristics of the 10,000 participants of the LIFE-ADULTStudy.**

|  | Total<br>n = 10.000 | Deceased by:<br>Dez. 2019/ Jan. 2020<br>n = 408 | Not deceased<br>n = 9592 | p-value |
|---|---|---|---|---|
| Age median (quartiles) | 57.9 (47.6; 68.2) | 70.0 (63.3; 74.9) | 57.3 (47.4; 67.7) | <0.001 |
| Sex (female) | 5197 (52%) | 132 (32%) | 5065 (53%) | <0.001 |
| Smokers | 2052 (22%) | 76 (22%) | 1976 (22%) | 0.925 |
| Body mass index (BMI) median (quartiles) | 26.6 (23.9; 30.0) | 28.1 (24.8; 31.7) | 26.5 (23.9; 29.9) | <0.001 |
| Myocardial infarction | 248 (3%) | 31 (8%) | 217 (2%) | <0.001 |
| Stroke | 219 (2%) | 27 (7%) | 192 (2%) | <0.001 |
| Hypertension | 4101 (44%) | 262 (69%) | 3839 (43%) | <0.001 |
| Hyperlipidaemia | 3100 (33%) | 160 (43%) | 2940 (33%) | <0.001 |
| Diabetes mellitus | 1073 (11%) | 117 (29%) | 956 (10%) | <0.001 |
| Asthma | 768 (8%) | 29 (7%) | 739 (8%) | 0.602 |
| Chronic bronchitis | 630 (7%) | 57 (15%) | 573 (6%) | <0.001 |
| Tuberkulosis | 226 (2%) | 26 (7%) | 200 (6%) | <0.001 |
| Liver cirrhosis | 40 (0.4%) | 9 (2%) | 31 (0.3%) | <0.001 |
| Peptic ulcer disease | 406 (4%) | 42 (11%) | 364 (4%) | <0.001 |
| Ulcerative colitis or Crohn's disease | 105 (1%) | 4 (1%) | 101 (1%) | 0.870 |
| Hepatitis | 833 (9%) | 44 (11%) | 789 (9%) | 0.074 |
| Renal insufficiency | 129 (1%) | 17 (4%) | 112 (1%) | <0.001 |
| Parkinson's disease | 29 (0.3%) | 6 (2%) | 23 (0.3%) | <0.001 |
| Epilepsie | 133 (1%) | 8 (2%) | 125 (1%) | 0.270 |
| Depression | 1058 (11%) | 29 (8%) | 1029 (11%) | 0.016 |
| Multiple sklerosis | 30 (0.3%) | 1 (0.3%) | 29 (0.3%) | 0.833 |
| Thyroid disease | 2647 (28%) | 90 (23%) | 2557 (28%) | 0.026 |
| Inflammatory rheumatic disease | 480 (5%) | 28 (7%) | 452 (5%) | 0.059 |
| Autoimmune diseases | 308 (3%) | 9 (2%) | 299 (3%) | 0.292 |
| Osteoporosis | 111 (1%) | 13 (3%) | 98 (1%) | <0.001 |
| Neurodermatitis | 369 (4%) | 10 (3%) | 359 (4%) | 0.168 |
| Cancer | 982 (10%) | 96 (25%) | 886 (10%) | <0.001 |
| Sepsis | 499 (5%) | 31 (8%) | 468 (5%) | 0.015 |
| Human immunodeficiency virus (HIV) | 12 (0.1%) | 0 (0%) | 12 (0.1%) | 0.473 |

Myocardial infarction was present in 248 (3%), stroke in 219 (2%), hypertension in 4101 (44%) and diabetes in 1073 (11%) subjects. 3100 (33%) subjects had hyperlipidaemia.

Total numbers and proportions of other diseases involving at least 5 percent of the cohort are: thyroid disorder: 2647 (28%), depression: 1058 (11%), cancer: 982 (10%), hepatitis: 833 (9%), asthma: 768 (8%), chronic bronchitis: 630 (7%) and sepsis 499 (5%). For the complete participant characteristic, see Table 1.

## Incidence of mortality

Within the limits of the request to the Saxon registration register (until December 2019), 408 deaths were reported. We were not allowed to evaluate the status of 62 participants due to the specific wishes for anonymity of these participants. The status of another 54 participants could not be determined because the available data was insufficient for the reliable identification of these participants. The number of living participants at the time of the request was 9475. The observed and expected numbers of deaths were compared. Fig 1 shows the cumulative deaths for both, the observed and expected numbers of deaths from 2011 to 2019. The expected probability of death is the calculation of how many participants would have died if the LIFE-ADULT cohort had been nationally representative. Hereby age, gender, and the fact that participants get older every year were taken into account. In total 895 deaths were expected, which is more than twice the number of deaths observed. The median observation time was 76.7 months (95% confidence interval: 75.9; 77.1).

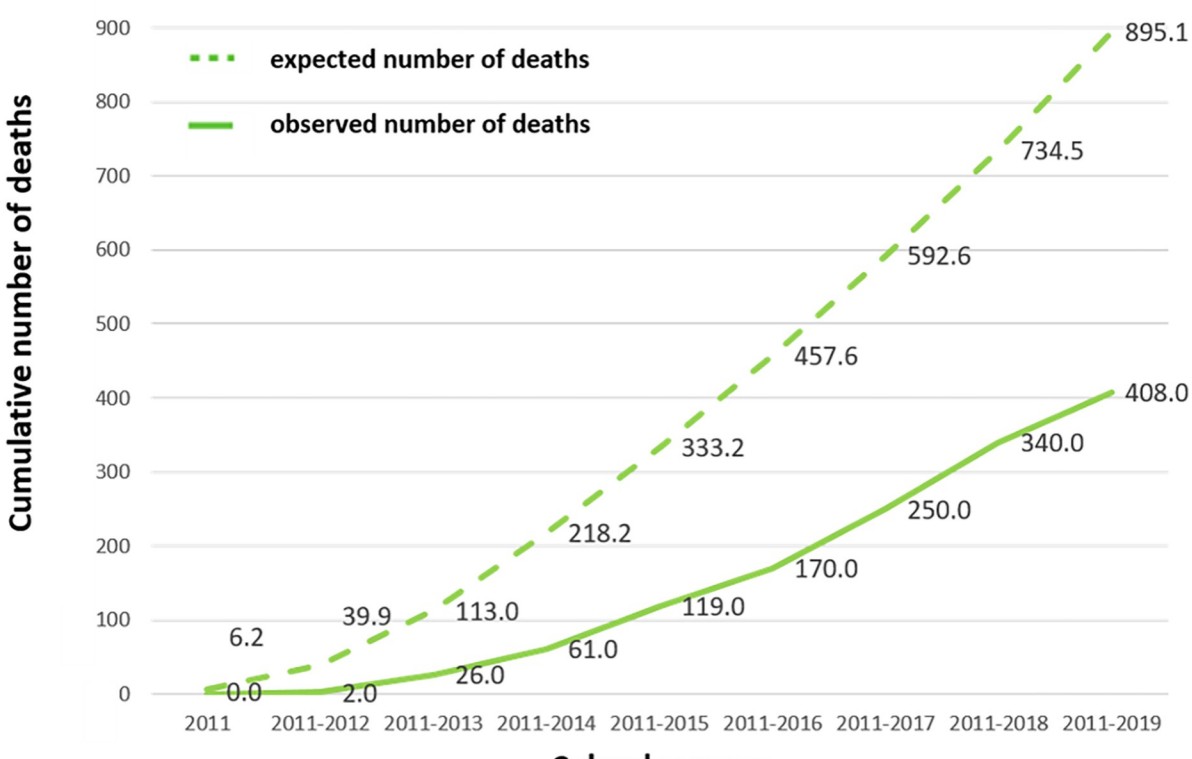

**Fig 1. Cumulative numbers of death.** Shown are the cumulative numbers of deaths. The solid line shows the observed numbers of deaths within the population investigated in the LIFE-ADULT-Study and the broken line the expected number of deaths for the LIFE-ADULT-Study. The expected number of deaths is calculated based on the life table information.

**Table 2. Absolute incidence of expected and observed deaths according to calender year.**

| Year | expected | observed |
|------|----------|----------|
| 2011 | 6.2 | 0 |
| 2012 | 33.7 | 2 |
| 2013 | 73.1 | 24 |
| 2014 | 105.2 | 35 |
| 2015 | 114.9 | 58 |
| 2016 | 124.5 | 51 |
| 2017 | 135.0 | 80 |
| 2018 | 141.9 | 90 |
| 2019 | 160.6 | 68 |

The absolute numbers of expected and observed deaths within one calender year are displayed in Table 2.

## Univariate analysis: OS for each comorbidity

In the univariate analysis myocardial infarction, coronary heart disease, stroke, hypertension, hyperlipidaemia, diabetes mellitus, chronic bronchitis, tuberkulosis, peptic ulcer disease, liver cirrhosis, renal insufficiency, Parkinson's disease, depression, thyroid disease, osteoporosis, cancer and sepsis were significantly associated with suvival time. For details on the relations of these diagnoses see Table 3.

## Univariate analysis adjusted for age and sex: Cox regression model for each comorbidity

After adjusting the previous univariate analysis for age and sex, survival time differences for the following 10 diagnoses persisted to show significance on a 5% level: myocardial infarction, stroke, hypertension, diabetes mellitus, chronic bronchitis, peptic ulcer disease, liver cirrhosis,

**Table 3. Relation between single diagnoses and survival time for diagnoses with significant relations according to univariate analysis (p-value $<$0.05).**

| Diagnosis | Hazard ratio | 95% confidence interval | p-value |
|-----------|--------------|-------------------------|---------|
| Myocardial infarction | 3.305 | 2.291, 4.767 | $< 0.001$ |
| Coronary heart disease | 2.701 | 1.899, 3.843 | $< 0.001$ |
| Stroke | 3.234 | 2.189, 4.779 | $< 0.001$ |
| Hypertension | 2.742 | 2.207, 3.407 | $< 0.001$ |
| Hyperlipidaemia | 1.462 | 1.192, 1.794 | $< 0.001$ |
| Diabetes mellitus | 3.370 | 2.718, 4.177 | $< 0.001$ |
| Chronic bronchitis | 2.396 | 1.809, 3.174 | $< 0.001$ |
| Tuberkulosis | 2.863 | 1.923, 4.263 | $< 0.001$ |
| Peptic ulcer disease | 2.689 | 1.952, 3.705 | $< 0.001$ |
| Liver cirrhosis | 6.390 | 3.299, 12.376 | $< 0.001$ |
| Renal insufficiency | 3.430 | 2.109, 5.577 | $< 0.001$ |
| Parkinson's disease | 5.206 | 2.324, 11.662 | $< 0.001$ |
| Depression | 0.622 | 0.426, 0.909 | 0.014 |
| Thyroid disease | 0.761 | 0.601, 0.964 | 0.023 |
| Osteoporosis | 2.885 | 1.660, 5.016 | $< 0.001$ |
| Cancer | 2.853 | 2.265, 3.593 | $< 0.001$ |
| Sepsis | 1.499 | 1.038, 2.164 | 0.031 |

**Table 4. Relation between single diagnoses and survival time, adjusted for age and sex.**

| Diagnosis | Hazard ratio | 95% confidence interval | p-value |
|---|---|---|---|
| Myocardial infarction | 1.487 | 1.025, 2.158 | 0.037 |
| Stroke | 1.804 | 1.217, 2.672 | 0.003 |
| Hypertension | 1.437 | 1.146, 1.801 | 0.002 |
| Diabetes mellitus | 1.860 | 1.493, 2.316 | < 0.001 |
| Chronic bronchitis | 1.808 | 1.363, 2.398 | < 0.001 |
| Peptic ulcer disease | 1.671 | 1.211, 2.308 | 0.002 |
| Liver cirrhosis | 5.192 | 2.680, 10.058 | < 0.001 |
| Renal insufficiency | 2.250 | 1.381, 3.664 | 0.001 |
| Parkinson's disease | 2.836 | 1.265, 6.357 | 0.011 |
| Osteoporosis | 1.938 | 1.101, 3.410 | 0.022 |
| Cancer | 1.700 | 1.342, 2.154 | < 0.001 |

renal insufficiency, Parkinson's disease, osteoporosis and cancer. For details on the relations of these diagnoses see Table 4.

## Multivariate analysis: Cox regression model for all reported comorbidities adjusted for age, sex, BMI and smoking status

In multivariate analysis adjusted for age, sex, BMI and smoking status, hypertension, diabetes mellitus, asthma, chronic bronchitis, peptic ulcer disease, Parkinson's disease, osteoporosis and cancer were significantly associated with the survival time (Table 5). Tuberculosis was not significantly but relevantly associated with survival, with p-value less than 10%.

## Optimal model: Cox regression model for all signifffikant and relevant comorbidities adjusted for age, sex, BMI and smoking status were included in this model

Based on consistency with clinical observations and a method of D. Collet [1], we submitted a selection of factors of the previous multivariate analysis to a subsequent multivariate analysis adjusted for age, sex, BMI > 25, and smoking status.

**Table 5. Relation between single diagnoses and survival time for diagnoses with significant relations according to multivariate analysis.**

| Diagnosis | Hazard ratio | 95% confidence interval | p-value |
|---|---|---|---|
| **p-value** < **0.05** (significantly associated): | | | |
| Hypertension | 1.500 | 1.139, 1.976 | 0.004 |
| Diabetes mellitus | 1.452 | 1.080, 1.952 | 0.014 |
| Asthma | 0.512 | 0.298, 0.880 | 0.015 |
| Chronic bronchitis | 2.337 | 1.650, 3.310 | < 0.001 |
| Peptic ulcer disease | 1.532 | 1.019, 2.304 | 0.040 |
| Parkinson's disease | 2.479 | 1.001, 6.137 | 0.050 |
| Osteoporosis | 2.367 | 1.291, 4.338 | 0.005 |
| Cancer | 1.959 | 1.486, 2.583 | < 0.001 |
| **p-value** < **0.1** (relevantly associated): | | | |
| Tuberculosis | 1.525 | 0.949; 2.449 | 0.081 |

Myocardial infarction, stroke, liver cirrhosis and renal insufficiency did not influence the overall survival.

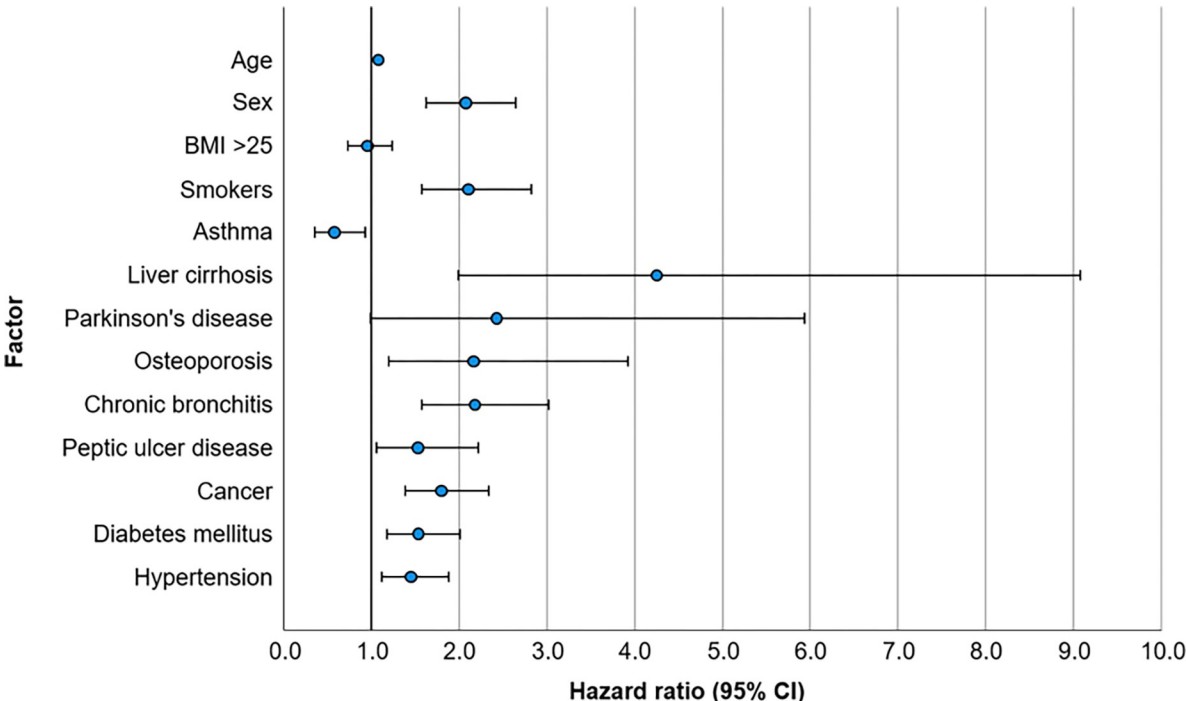

**Fig 2. Comparison of risks for mortality for risk factors and selected diagnoses, according to the optimal model.** Shown are the hazard ratios and 95% confidence intervals for the risk factors age, sex, BMI > 25 and smoking status and several diagnoses in the medical history of the participants of the LIFE-ADULT-Study, selected by the optimal model.

This resulted in an optimal model including the following diseases/factors: asthma, liver cirrhosis, Parkinson's disease, osteoporosis, chronic bronchitis, peptic ulcer disease, cancer, diabetes mellitus and hypertension.

With exception of the diagnosis Parkinson's disease survival time differences showed significance on a 5% level for the factors constituting the optimal model (Fig 2, Table 6).

## Discussion

In the population-based LIFE-Adult-Study, the overall mortality rate was nearly about half the rate of the german population. Adjusting for age, sex, BMI and the smoking status, liver

**Table 6. Relation between diagnoses and survival time for diagnoses selected by the optimal model.**

| Diagnosis | Hazard ratio | 95% confidence interval | p-value |
|---|---|---|---|
| **p-value < 0.05:** | | | |
| Asthma | 0.574 | 0.355, 0.928 | 0.024 |
| Liver cirrhosis | 4.251 | 1.990, 9.082 | < 0.001 |
| Osteoporosis | 2.165 | 1.195, 3.925 | 0.011 |
| Chronic bronchitis | 2.179 | 1.574, 3.017 | < 0.001 |
| Peptic ulcer disease | 1.531 | 1.057, 2.219 | 0.024 |
| Cancer | 1.797 | 1.382, 2.337 | < 0.001 |
| Diabetes mellitus | 1.533 | 1.173, 2.004 | 0.002 |
| Hypertension | 1.447 | 1.116, 1.876 | 0.005 |
| **p-value < 0.1:** | | | |
| Parkinson's disease | 2.425 | 0.992, 5.933 | 0.052 |

cirrhosis, osteoporosis, chronic bronchitis, peptic ulcer disease, cancer, diabetes mellitus and hypertension had significant impact on the overall survival in multivariate analyses by increasing the hazard ratio whereas asthma had significant impact on the overall survival by decreasing the hazard ratio. To our knowledge, information about mortality rates and event dependent overall survival in a population-based cohort study compared to the german general population is not reported yet.

## Mortality

Our findings are consistent with other epidemiological studies when comparing mortality rates between large population based studies and the general population [8–10]. A plausible and already accpeted explanation for this effect is that the participation rate of epidemiological studies is never 100% and therefore there is always a selection bias of the participants. The participation rate of 33% we registered here is low but comparable to the extent of participation in other large cohort studies [11–13]. The problem of the relatively low participation rate is the selection bias due to the fact that participation in epidemiological studies depends on social, educational and health conditions [9,14,15]. This has already been discussed in connection with several other epidemiological studies, e.g. the "Diet, Cancer and Health" study in Denmark (Larsen SB 2012). As recently reported, low baseline participation in the LIFE-Adult-Study was associated with the typical selection of study participants with higher social status and healthier lifestyle, and additionally less concomitant diseases [16]. Since only about thirty percent of the randomly selected persons participated in the LIFE-Adult baseline examination the analesed cohort is not entirely representative. Enzenbach et al. compared LIFE-ADULT participants aged 40–79 years with the Leipzig population and non-participants, detecting considerable differences in education and health variables as well as identifying a lack of time and interest and health problems as the main reasons for nonparticipation [16].

Some of the non-participants (about 32%) filled a short questionnaire. LIFE-Adult participants reported also less often to have been diagnosed with myocardial infarction, diabetes and stroke as nonparticipants.

We think that these differences in participants and non-participants are reflected by the low mortality rate observed here. Consequently, the main health conditions will be underestimated in LIFE-Adult compared to the general Leipzig population. In the multivariate Cox regression-models for overall survival we adjusted all diagnoses reported to us with some risk factors as age, gender, BMI, and smoking status.

In many epidemiological studies loss to follow-up is also problematic and often leads to bias. Because cohort studies typically take several years to follow up participants, it is expected that some of the participants will be lost with time. If some individuals with the same risk factors are lost from the same single high-risk groups, this can lead to a bias of poor differential classification by risk. The advantage of our work in the LIFE-ADULT-Study is that access to mortality via the reporting register provides reliable and complete data, independent of the further contribution of the participants. Therefore, we were able to conduct our analysis for nearly the entire LIFE-ADULT cohort (98.7%).

## Comparison of risk factors

In the year 2018 18.5% of adult women and 24.2% of adult men were smoking [17]. This is comparable to the numbers observed in the LIFE-ADULT-Study. The median BMI of the participants in the LIFE-ADULT-Study was slightly higher (26.6 kg/m$^2$) than the median BMI of 26.2 kg/m$^2$ in men and 24.3 kg/m$^2$ in women in the cohort evaluated in the German National Nutrition Survey II [18]. The median age in the LIFE-ADULT cohort was higher (57.9 years)

than in the German National Nutrition Survey II (49.0 years for men and 48.0 years for women). As BMI tends to increase with age we still can consider the LIFE-ADULT cohort to be representative for the german population regarding the BMI.

## Comparison of medical history and impact on mortality

The mortality rate of the LIFE-Adult cohort was significantly lower compared to the Leipzig general population and to non-responders. To evaluate if there is a relevant interaction between several diseases of the medical history and the overall mortality, we compared the prevalence of several diseases occurring in the LIFE-ADULT cohort with the findings from the DEGS1 (Studie zur Gesundheit Erwachsener in Deutschland) cohort [19]. Regarding stroke prevalence, the findings were comparable [20]. In the LIFE-ADULT-Study we observed that 2.3% of the subjects had stroke in their medical history compared to 2.9% in the DEGS1 study. Different results were present for the prevalence of myocardial infarction [21]. In the LIFE-A-DULT-Study, 2.7% of myocardial infarctions were reported, while 4.7% were found in the DEGS1 study. This difference might be due to the different age distributions of both studies. We examined data from participants aged 18–80 years, while the study population examined in the DEGS1 publication on myocardial infarction was 40–79 years old and the prevalence for myocardial infarction increases with age [22]. The cases of Diabetes mellitus reported by the participants of the LIFE-ADULT-Study, on the other hand, are slightly higher (10.7%) than the number of patients with diabetes in Germany (10.1%), evaluated from the insurance data of the entire German population, from the age of 0 up [23]. As the occurance of diabetes also increases with age, this very small difference might again be due to the age difference in the two cohorts. 44.1% of the participants in the LIFE-ADULT-Study reported hypertension. In the DEGS1 study, the total proportion of participants with known hypertension was 25.9%, and hypertensive blood pressure values were measured in a further 5.6% [24]. The participants in the DEGS1 study were also between 18 and 79 years old, but the age distribution may be different. The average age of the participants in the LIFE-ADULT-Study is 57.9 years. In the age group 50–59 years, the proportion of participants with known hypertension in the DEGS1 study is 30.4%, while in the age group 60–69 years it is already 53.4%. The prevalence of hypertension increases steeply with age and therefore the age distribution of the LIFE-ADULT--Study could cause the observed differences in the number of participants with hypertension. In contrast to hypertension the numbers for depression are lower in the LIFE-ADULT cohort. In the LIFE-ADULT-Study 10.6% of participants reported depression while a total proportion of 15.7% persons with depression was reported in 2017 in Germany [25]. This also is observed for asthma. The lifetime prevalence for developing asthma according to the DEGS1 study is slightly higher (8.6%) [26] than the proportion of participipants reporting asthma in their medical history in the LIFE-ADULT-Study. Differences were also found for cancer: 9.8% of the participants of the LIFE-ADULT-Study reported cancer in their medical history. The risk of developing cancer before the age of 75 years is 30.1% for the german population [27]. Although it is difficult to compare these numbers as the median age of the population in the LIFE-ADULT-Study is 57.9 years and the risk to develop cancer increases with age, the reported number of participants with a history of cancer is quite low. Based on these examples of common diagnoses, it can therefore be said that the cohort examined in the LIFE-ADULT--Study is only partially representative of the German population. The observed differences in the mortality rates might therefore be explained by differences in the occurance of diagnoses in the medical history of the participants.

The comparative analyses of the UK Biobank study (a big British cohort study, having a very low response proportion of only 5.5%), still show that the general cause-effect

relationships should be unaffected by the effects of low response proportions and apply to the population of the wider survey area [9]. Accordingly we assume that concerning cause-effect relationships, our sample should be representative for the german population.

This study showed the significantly increased mortality risk if the following diseases are reported in medical history of the participants in a large *population*-based *cohort* study including *adults aged* 18 and *over*: diabetes mellitus, hypertension, liver cirrhosis, osteoporosis, chronic bronchitis, peptic ulcer disease and cancer. The impact of these diagnoses proved to be significant after adjusting for the known risk factors age, sex, body mass index and smoking status. The initial selection of diagnoses comprises common diseases influencing mortality or diseases with pathomechanisms allowing to assume possible ties to mortality. Furthermore, the choice was limited to conditions asked for in the framework of the LIFE-ADULT-Study and having case numbers high enough for building stable statistical models.

The diagnosis of diabetes mellitus refers to a group of diseases that are characterized by elevated blood sugar levels. These higher glucose concentrations adversely affect the cardiovascular system through uncontrolled glycation of structural and functional proteins and lipids [28]. There is a strong hereditary component to the risk of developing both type 1 and type 2 diabetes. Type 2 diabetes, which is responsible for 90% of cases, is also significantly associated with the metabolic syndrome [29]. Diabetes itself is a significant risk factor for the development of micro- and macroangiopathies affecting the cardiovascular and nervous systems, the kidneys and damage the eyes [30]. Implicitly, diabetes is influenced by BMI and hypertension, while self-identifying as a significant risk for stroke, myocardial infarction and renal failure. Finally, both diabetes and arterial hypertension impair the function of numerous organ systems by promoting atherosclerosis of the small and large vessels that supply the respective organs, resulting in fatal events including stroke, myocardial infarction, congestive heart failure, aortic rupture or dissection, and renal failure.

Regardless of the specific etiology, cirrhosis of the liver is a liver damage caused by liver fibrosis that is irreversible at advanced stages [31]. Impaired synthesis of coagulation factors and immunoglobulins, reduced metabolic capacity and loss of physiological vascularization causes bleeding tendencies, immunodeficiency, hepatic encephalopathy, portal hypertension and ascites [32]. As a result, patients have an increased risk of variceal bleeding, pneumonia, spontaneous bacterial peritonitis, lung and kidney failure and even multi-organ failure in the event of acute decompensation. Last but not least, cirrhosis is a major cause of hepatocellular carcinomas [33].

Osteoporosis is characterized by low density and structural deterioration of the bones, exposing the patient to an increased risk of fracture and, consequently, mortality from risks inherent in the surgical procedure and complications such as infection. For example, the one-year mortality rate from a hip fracture, a classic type of osteoporosis-related fracture, is reported to be more than 20% [34].

Peptic ulcer disease can lead to hemorrhagic shock from gastrointestinal bleeding and peritonitis or even sepsis after perforation [35]. Also, the affinity of ulcerative tissue defects for cancerous degeneration predisposes patients to the development of gastric cancer [36].

Chronic bronchitis is a phenomenon in chronic obstructive pulmonary disease (COPD) characterized by overproduction and hypersecretion of mucus. Clinical consequences can include accelerated decline in lung function, development of airway obstruction, lower respiratory tract infections, and worsened all-cause mortality [37]. The main cause for chronic bronchitis is smoking. Although adjusting for this risk factor, chronic bronchitis remained significant in the optimal model.

Asthma also significantly affected the risk of death, but not by increasing but by reducing the risk of death. Asthma is a chronic inflammatory disease of the lower airways with a

heterogenous etiology and a wide range of severity. The severity can range from very mild seasonal allergic reactions to severe conditions affecting every day life and mobility by causing diminished lung function [38]. The reported proportion of patients with severe asthma ranges from 4.2% [39] to more than 30% [40]. In the LIFE-ADULT-Study no information regarding the severity was collected but having the selection bias for epidemiological studies in mind the proportion of participants with severe asthma might be underrepresented in the LIFE-ADULT-Study. The treatment of mild and moderate asthma includes regular physical exercise, smoking cessation and avoidance of passive smoke exposure and an optimization of body weight [41]. We can hypothesize that the combination of a high proportion of participants with mild and moderate asthma and the education of these patients to a healthy lifestyle might cause the reduced mortality riks for participants with asthma in their medical history.

## Discussing the absence of known risk factors like stroke and myocardial infarction in the final model

Looking at the results, it is striking that two of the most common diseases associated with mortality in industrialized nations, myocardial infarction and stroke, did not make it into the final selection of the analysed diseases. Myocardial infarction and stroke can be viewed as effectors manifesting the deleterious influence of their underlying causes such as diabetes, hypertension, aging, BMI and smoking. Given the cause-and-effect relationship, we should expect a relatively large overlap of these factors among participants, leading to a split in effect size. Furthermore, the splitting should be asymmetrical in nature, to the detriment of effectors (or dependents) such as stroke and myocardial infarction, due to the following consideration in this example:

The group of deceased participants with a stroke event will, of course, contain many participants who also have one or more of the above risk factors for stroke, hence the effect size of stroke is reduced by each associated risk factor in the analysis. Conversely, each individual risk factor suffers only a fraction of the reduction because the overlap with a history of stroke is shared between the different risk factors in the analysis. Second, it should be noted that the study cannot account for the contribution to mortality from participants with a matched disease but who have already died from the same disease. Ergo, there is an inherent study selection towards patients with less severe events, simply because patients who died from disease-related fatal events cannot participate.

Considering, for example, that mortality from the first stroke is almost 20%, this selection effect should significantly shift the cohort toward lower mortality, particularly for diagnoses that also equate to potentially fatal events, such as stroke and myocardial infarction.

## General limitation of the analysis

Although collecting data in epidemiological studies via the self-report mode is a well-established method, it is necessary to pay attention to apparent inherent flaws that exist and the extent of their impact on our survey.

For example, both the frequent use of technical terms (such as "cerebral infarction" vs. "stroke") and the use of well-intentioned simplifications (such as "weakness of the heart") are likely to encourage multiple misstatements in self-reports. Also the lack of medical understanding can cause wrong information. "Infarction" can be considered "myocardial infarction" because the term "cerebral infarction" is not known, "heart failure" can be interpreted as "heart attack" instead of the correct "heart failure", the "transient ischemic attack" can be recorded as a "stroke" or omitted entirely.

In addition, there appear to be differences in the reliability of self-reported illness, which depends on the specific illnesses reported. One might expect that chronic conditions would be

more reliable than individual events such as a minor stroke, but even comparing the reliability of multiple chronic conditions reveals significant differences. For example, in a large cohort study, the European Prospective Investigation into Cancer and Nutrition, positive predictive values for self-reported diseases range from relatively high validity for arterial hypertension 79.4% [42] and diabetes 87.8% (Sluijs I 2010) to intermediate values of 60.7% for myocardial infarction down to a meager 22.2% for stroke [43].

In general, when discussing the results, it must be taken into account that the risk indicators are distorted both by the selection of the risk factors that were used for adjusting and by the overall selection of the diseases included.

For example, although we did not adjust the analysis for other factors such as alcohol consumption or lack of exercise because the relevant data were not available or not detailed enough, it is important to bear this in mind when looking at the results—the selective adjustment shifts the risk proportions towards relatively higher risks for diagnoses that would have been influenced by these missing factors.

For example, adjusting for alcohol consumption would undoubtedly have affected the resultant risk of liver cirrhosis, since it is known to be a significant risk of liver cirrhosis (Rehm J 2010).

In addition to the interaction with adjusted risk factors, diagnoses can show complex cause-effect relationships among themselves. It should then be helpful to also discuss pathomechanisms, individual risk factors and whether the disease itself represents a risk factor for other diseases.

## Conclusion

The present study confirms the selection effect of epidemiological studies towards a healthier population, which results in reduced mortality. Nevertheless, diseases associated with increased mortality could be identified. This confirmed that, despite the selection of the study population, the basic cause-effect principles also apply to the study population of the LIFE-ADULT-Study.

## Acknowledgments

Leipzig Research Centre for Civilization Diseases (LIFE) is an organizational unit affiliated to the Medical Faculty of the University Leipzig. The authors thank the participants of the LIFE-Adult-Study for taking part in the study and the complete LIFE-Adult team for organizing the course of the study and performing the examinations.

We thank Ulrike Schoenwiese und Ute Enders for technical assistance and search in the registry.

## Author Contributions

**Conceptualization:** Samira Zeynalova, Markus Loeffler.

**Formal analysis:** Samira Zeynalova.

**Investigation:** Michael Brosig.

**Software:** Matthias Reusche.

**Writing – original draft:** Samira Zeynalova, Katja Rillich, Eike Linnebank.

**Writing – review & editing:** Samira Zeynalova, Katja Rillich, Tina Stegmann, Michael Brosig, Matthias Reusche, Markus Loeffler.

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
