## [Decision Letter · Decision Letter 0]

9 Aug 2022

PONE-D-22-10390Event dependent overall survival in the population-based LIFE-Adult-Study

PLOS ONE

Dear Dr. Rillich,

Thank you for submitting your manuscript to PLOS ONE. After careful consideration, we feel that it has merit but does not fully meet PLOS ONE’s publication criteria as it currently stands. Therefore, we invite you to submit a revised version of the manuscript that addresses the points raised during the review process.

A marked-up copy of your manuscript that highlights changes made to the original version. You should upload this as a separate file labeled 'Revised Manuscript with Track Changes'.An unmarked version of your revised paper without tracked changes. You should upload this as a separate file labeled 'Manuscript'.

We look forward to receiving your revised manuscript.

Kind regards,

Steve Zimmerman, PhD

Associate Editor, PLOS ONE

Journal Requirements:

"Leipzig Research Centre for Civilization Diseases (LIFE) is an organizational unit affiliated to the Medical Faculty of the University Leipzig. LIFE was funded by means of the European Union, by the European Regional Development Fund (ERDF) and by funds of the Free State of Saxony within the framework of the excellence initiative (project numbers 713-241202, 713- 241202, 14505/2470, 14575/2470). The authors thank the participants of the LIFE-Adult-Study for taking part in the study and the complete LIFE-Adult team for organizing the course of the study and performing the examinations. Data evaluation for this publication was supported by Sanofi Genzyme, Germany, by a non-restricted scientific grant. 

We thank Ulrike Schoenwiese und Ute Enders for technical assistance and search in the registry."

Reviewers' comments:

Reviewer's Responses to Questions

**Comments to the Author**

1. Is the manuscript technically sound, and do the data support the conclusions?

Reviewer #1: Yes

Reviewer #2: Yes

2. Has the statistical analysis been performed appropriately and rigorously? 

Reviewer #1: Yes

Reviewer #2: Yes

3. Have the authors made all data underlying the findings in their manuscript fully available?

Reviewer #1: No

Reviewer #2: Yes

4. Is the manuscript presented in an intelligible fashion and written in standard English?

Reviewer #1: Yes

Reviewer #2: Yes

5. Review Comments to the Author

Reviewer #1: Dear authors, thank you for giving me the opportunity to review this manuscript. This research has an excellent rigor and provide some essential contributions to the field.

On the other hand the authors should consider some improvements for their review.

Line 112-131 it is not clear, there the information of cause of death came from. Are these diagnosis from the LIFE study, or the information from the death certificates. However the authors should explain if causes of death information is used. Are these underlying causes, any diagnosis or something else?

Otherwise the authors should make some explanations for specific diagnosis. Osteoporosis is not a common underlying cause. It will be used as proxy for specific risks like falls.

Also individuals with osteoporosis generally have higher mortality risk, because it is more likely in age grougs 80 and 90 +. These comparisions are more plausible if comorbidities are considered. Please look at the paper of Ensrud K, Kats A, Boyd C, et al. Association of disease definition, comorbidity burden, and prognosis with hip fracture probability among late-life women in JAMA Intern Med.

Renal insufficieny is also a secondary diagnosis.

Consequently the authors should define between underlying causes/major diagnosis and secondary diagnosis.

Generally the authors should used a delayed-entry model, while LiFE individuals are entering the study population at different time points. So they have differtent individual and age-specific mortality risk before and after entering LIFE

The authors should also compensate the number of models. For the reader it is not almost comprehensive to interpret the findings from various models and tables.

For their revision, authors should consider all these comments.

Reviewer #2: Event dependent overall survival in the population-based LIFE-Adult-Study

PONE-D-22-10390

Observations

It was interesting reading the article, not just because such studies are rare in developing

countries but also because it has disease diagnosis and risk factors in the data set. The study

demonstrates the role of the disease in the survival time of the adult after controlling the

risk factors. The authors have calculated survival time and cumulative incidence of mortality

from August 2011 until the end of 2019 (line no. 102-104) using the time of the occurrence

of the death out of the cohort of 10,000 adults aged 18-80 years in Leipzig, Germany.

The authors of the paper have attempted and also discussed all the limitations of the cohort

study and elaborated on why there is a difference in the mortality of the members of the

cohort and the general population (Line no. 32-34). I see this as the weakest point of this

paper, as the study findings can not be generalized to the general population of Germany.

This, in no way, undermines the value of the results as it may apply to the specific

population of a particular geographic area. I, however, have the following specific

observations on the present paper:

1. Line no. 72-75- The authors mention that the study subjects are randomly selected

stratified by age and sex. The question that immediately comes to me is if this is so,

then why is there a huge difference in mortality of the cohort and general

population?

2. There are two main explanations given: one low participation (33%, Line no. 237-

238) and another one ---selection of study population with higher social status and

healthier lifestyle—---(Line No. 244-245). The question then is, “why such

differences in the population Characteristics”? In my view, it may be that study is

located in a particular geographic location with economically better-off people and a

good lifestyle. The authors must provide explanations for such differences. I did not

find anything on the geographic spread of the study population. The authors need to

include them in the paper.

3. Line No. 370-387: The last point again is about the natural selection bias. While

trying to explain the absence of some known diseases (Myocardial infarction and

stroke) from the list of the cause of death, authors rely on overlapping effects and

interaction of some causes of death, and therefore the size of contribution gets

reduced. This may be true, but the second explanation needs some evidence. Paper

fails to provide any data for this.

It is hypothesized that those with such severe causes might have died and therefore

have no chance of selection in the study. This seems to me less likely in the

population where all-cause mortality is so low, and the age group for the study is 18-

80 years. If possible, some evidence such as selection bias even from the review may

be included.

The overall study contributes to our understanding of survival chances among those who

are diagnosed with some chronic diseases. Therefore, it is worth publishing after some more

explanation, as pointed out in these observations.

Usha Ram

6. PLOS authors have the option to publish the peer review history of their article (what does this mean?). If published, this will include your full peer review and any attached files.

Reviewer #1: **Yes: **Ronny Westerman

Reviewer #2: **Yes: **Usha Ram

---

## [Author Response · Author response to Decision Letter 0]

21 Oct 2022

We have tried to answer all questions and consider all suggestions for improvement. You can find our answers in the file "Response to Reviewers". 

With best regards,

Samira Zeynalova and Katja Rillich

---

## [Editor Report · Decision Letter 1]

9 Nov 2022

Event dependent overall survival in the population-based LIFE-Adult-Study

PONE-D-22-10390R1

Dear Dr. Zeynalova,

We’re pleased to inform you that your manuscript has been judged scientifically suitable for publication and will be formally accepted for publication once it meets all outstanding technical requirements.

Kind regards,

Usha Ram

Guest Editor

PLOS ONE
---

## [Editor Report · Acceptance letter]

17 Nov 2022

PONE-D-22-10390R1 

Event dependent overall survival in the population-based LIFE-Adult-Study 

Dear Dr. Rillich:

I'm pleased to inform you that your manuscript has been deemed suitable for publication in PLOS ONE. Congratulations! Your manuscript is now with our production department. 

Kind regards, 

on behalf of

Usha Ram 

Guest Editor

PLOS ONE